# Improving Patient Safety in the X-ray Inspection Process with EfficientNet-Based Medical Assistance System

**DOI:** 10.3390/healthcare11142068

**Published:** 2023-07-19

**Authors:** Shyh-Wei Chen, Jyun-Kai Chen, Yu-Heng Hsieh, Wen-Hsien Chen, Ying-Hsiang Liao, You-Cheng Lin, Ming-Chih Chen, Ching-Tsorng Tsai, Jyh-Wen Chai, Shyan-Ming Yuan

**Affiliations:** 1Department of Computer Science, Tunghai University, Taichung 407224, Taiwan; chensw@thu.edu.tw (S.-W.C.); cttsai@thu.edu.tw (C.-T.T.); 2Department of Computer Science, National Yang Ming Chiao Tung University, Hsinchu 300093, Taiwan; jyunkaichen.cs10@nycu.edu.tw (J.-K.C.); k28998989.cs11@nycu.edu.tw (Y.-H.H.); 3Department of Radiology, Taichung Veterans General Hospital, Taichung 407219, Taiwan; stayaway@vghtc.gov.tw (Y.-H.L.); clusslin@vghtc.gov.tw (Y.-C.L.); ku4199@vghtc.gov.tw (M.-C.C.); hubt@vghtc.gov.tw (J.-W.C.); 4Department of Industrial Engineering and Enterprise Information, Tunghai University, Taichung 407224, Taiwan; 5Department of Post-Baccalaureate Medicine, College of Medicine, National Chung Hsing University, Taichung 402202, Taiwan; 6Post-Baccalaureate Medicine, National Chung Hsing University, Taichung 402202, Taiwan; 7College of Medicine, China Medical University, Taichung 406040, Taiwan

**Keywords:** deep learning, Artificial Intelligence (AI), medical process, Convolutional Neural Network (CNN), image classification, error detection, early warning, patient safety

## Abstract

Patient safety is a paramount concern in the medical field, and advancements in deep learning and Artificial Intelligence (AI) have opened up new possibilities for improving healthcare practices. While AI has shown promise in assisting doctors with early symptom detection from medical images, there is a critical need to prioritize patient safety by enhancing existing processes. To enhance patient safety, this study focuses on improving the medical operation process during X-ray examinations. In this study, we utilize EfficientNet for classifying the 49 categories of pre-X-ray images. To enhance the accuracy even further, we introduce two novel Neural Network architectures. The classification results are then compared with the doctor’s order to ensure consistency and minimize discrepancies. To evaluate the effectiveness of the proposed models, a comprehensive dataset comprising 49 different categories and over 12,000 training and testing sheets was collected from Taichung Veterans General Hospital. The research demonstrates a significant improvement in accuracy, surpassing a 4% enhancement compared to previous studies.

## 1. Introduction

According to the World Health Organization’s (WHO) fact sheet on patient safety [1] and the latest TPR 2020 Annual Report [2], a significant number of patients are at risk of experiencing harm due to incorrect medical management [3]. Nevertheless, it is worth noting that close to 50% of these adverse events are preventable [4]. Misidentification of the patient’s examination site is a major source of error events in hospital radiology departments, as shown in various examples such as those in Figure 1 and Figure 2. For instance, a doctor may order a right (R’t) WRIST anterioposterior (AP) image, but the radiographer may mistakenly take a R’t WRIST lateral (LAT) image, or an order of left (L’t) FOOT AP image may be taken as the L’t FOOT AP image, which potentially leads to incorrect diagnosis and treatment. Studies conducted by Sadigh et al. [5] at two large US academic hospitals showed that out of 1,717,713 examinations performed during their study period, 67 error reports were identified, with an estimated event rate of 4 per 100,000 examinations. Although the probability of such errors seems low, any errors can have serious consequences, leading to delayed diagnosis and treatment for patients.

Patient safety is of utmost importance in medical services, and any measures to prevent such errors should be taken. For example, the outpatient X-ray room of Taichung Veterans General Hospital performs nearly 1000 X-ray images daily, and in such a high-pressure environment, human errors are unavoidable. Therefore, it is crucial to improve medical procedures to address this issue and ensure patient safety.

Since 2013, there has been a significant rise in research focused on machine learning within the health and life sciences domain. Among the most widely adopted applications is the support it provides to doctors in diagnosing and treating patients. Despite the continued reliance of most hospitals on traditional methods, such as PDCA (Plan-Do-Check-Act), for enhancing medical procedures, researchers have introduced groundbreaking solutions [6,7,8,9,10] that utilize image classification to assist in diagnostics. For instance, Gao [7] employed a deep learning approach to classify Fundus Fluorescein Angiography (FFA) images, determining the presence or absence of diabetic retinopathy. Gupta [8] utilized two models, MobileNetV2 and DarkNet19, for classifying patients as either having or not having COVID-19. In [9], Pan et al. employed a deep learning model to categorize fundus images into three groups: normal, macular degeneration and tessellated fundus. The iERM system proposed by Kai et al. [10] is a two-stage Deep Learning system that enhances the grading performance and interpretability of ERM by incorporating human segmentation of key features.

Image classification is a vital task in computer vision, where CNNs play a crucial role [11,12,13,14,15,16,17,18,19,20,21,22,23,24]. They excel at extracting features and enabling accurate classification. CNNs are widely used and are particularly valuable in medical diagnostics. Transfer learning, using pre-trained models, such as ImageNet, is a common technique to enhance performance on new tasks with smaller datasets. Feature Extraction and Fine Tuning are the two main approaches to transfer learning. For this study, the PyTorch Image Models (timm) library by Ross Wightman [25] is used for feature extraction and the fine-tuning of medical images.

The outpatient X-ray room at Taichung Veterans General Hospital handles an average monthly workload of approximately 30,000 X-ray images, which translates to almost 1000 X-ray images generated daily. This workload is managed by only 3–4 radiologists and 1 support personnel, leading to a high-pressure work environment where human errors are likely to occur. After analyzing the causes of errors, including subjective and objective factors such as physical discomfort, unfamiliar inspection sites, long working hours, and high pressure, it was determined that corrections and assistance in the medical process could reduce such errors.

Miao [6] introduced a solution to tackle the mentioned problem, with the goal of classifying 49 categories of X-ray images. These categories encompass 25 sites, 40 categories with direction, and 9 categories without direction. The best result obtained was a testing accuracy of 94.10% trained with Xception [26]. However, Miao identified two issues that led to low accuracy: the feature gap between positive and lateral positions was too small and left and right images were difficult to classify. To enhance the accuracy of Miao [6], we used EfficientNet as our CNN model. We also propose two enhanced model architectures: T40P3x4 and T40P2x4A2P2 and implement three optimization strategies to address the issues of low accuracy: using a more robust model, data purification, and data augmentation. By implementing these improvements, the overall system accuracy increased by more than 4.0%, reaching 98.16%. This significant improvement has not only enhanced the quality of medical services but also improved patient safety.

The main contribution of this paper includes:According to TPR 2020 [2], it is evident that there is a higher likelihood of radiologists causing delays in patient diagnoses. However, the effective utilization of EfficientNet has significantly reduced human errors among radiologists. Compared to previous studies [6], our model demonstrates improved accuracy, and in addition to that, we offer F1-score, Recall, and Precision measurements;The 49-category classifier exhibits some misclassifications in certain body parts. To enhance the accuracy of these specific body parts, we propose the two-level classifier T40P3x4. This new approach promises to further improve the overall accuracy to 98%;Despite achieving 98% accuracy, the two-level architecture does encounter misclassifications in three body parts of RGB pictures. In order to address this limitation, we present a novel three-level architecture: T40P2x4A2P2. The latest methodology not only boosts the overall accuracy significantly but also efficiently classifies body parts that were previously misclassified using the two-level architecture, leading to a remarkable 98.16% accuracy.

The rest of the paper is organized as follows: In Section 2, we will introduce the datasets, materials and methods used in our study. Section 3 presents the results of our experiments. We will discuss the results, compare them with Miao [6] and talk about future work in Section 4. Finally, in Section 5, we draw our conclusions.

## 2. Materials and Methods

### 2.1. Data Collection

#### 2.1.1. Overview

This study was conducted using the X-ray room located in the outpatient department of Taichung Veterans General Hospital as the primary source of data. The data collection period lasted for 8 months, from September 2021 to April 2022. The images obtained in this study were of the most frequently examined areas, which constituted 80% of the daily workload for radiologists. The 15 examined areas were divided into 40 categories with directional indicators and 9 categories with no directional indicators, resulting in a total of 49 categories. In Figure 3 and Figure 4, the following abbreviations were used: AP and PA indicate anterioposterior and posterioanterior directions, LAT refers to lateral direction, OBL refers to an oblique direction, STD indicates standing position, L’t represents left, R’t represents right, C-spine represents cervical spine, T-spine represents thoracic spine and L-spine represents lumbar spine.

#### 2.1.2. Purification

During the data collection process, it was observed that a small number of images for some sites were different from the majority of images due to special conditions. For instance, in Figure 5, the patient’s position cannot be discerned from the image because of factors such as wearing a cast, a brace, or clothing covering the body. Additionally, in Figure 6, the radiation of the same site can vary significantly between children and adults or due to certain actions that the patient may be required to perform by the physician. Both of these scenarios can affect the accuracy of the data set and confuse the model’s recognition of categories during training. As a result, these special cases were removed in this study to avoid negatively impacting the model’s training.

This strategy is also expected to enhance the model’s ability to extract precise features for each site on the left and right, thereby improving the challenge of classifying left and right, as mentioned in Miao’s paper [6]. Since the number of X-ray types taken by the radiology department varied daily, this study selected 100–300 images for each category after data purification, resulting in a total of 12,152 images. Table 1 shows the number of RGB pictures for each body category. In these 20 body categories, we have highlighted the 11 directional body categories. We further break down the directional body categories into 40 additional categories in addition to the original 9 non-directional body categories. Figure 7 illustrates the data distribution across all these 49 categories. We observe that FEMUR LEG and ELBOW have a relatively lower number of RGB pictures, which could potentially lead to misclassification by the classifier. To tackle this issue of limited dataset size, we will employ a data augmentation scheme, as detailed in Section 2.2.2.

### 2.2. Methods

#### 2.2.1. System Workflow

The system workflow, as illustrated in Figure 8, begins with the system capturing an RGB picture prior to the radiologist performing an X-ray. This picture serves as the input to the classifier introduced in this paper, which aims to identify the specific body part being imaged.

The classifier analyzes the image and produces its classification results, indicating the identified body part. These results are then compared with the doctor’s order, which specifies the expected body part for the X-ray.

In the case where the classifier’s results align with the doctor’s order, indicating a correct identification of the body part, the system generates a notification to inform the radiologist that the process is complete. This notification serves as confirmation that the X-ray is ready to be taken.

However, if there is a mismatch between the classifier’s results and the doctor’s order, suggesting a potential incorrect identification of the body part, the system generates a warning notification. This notification is sent to alert the radiologist of the discrepancy, prompting further investigation and ensuring the correct body part is imaged before proceeding with the X-ray.

By implementing this workflow, the system enhances the accuracy and reliability of body part identification during X-ray procedures, providing valuable support to radiologists and reducing the risk of misdiagnosis or procedural errors.

#### 2.2.2. Data Augmentation

In order to overcome the issue of insufficient data diversity resulting from the scarcity of medical images, this study has implemented data augmentation in addition to transfer learning. Figure 9 demonstrates four general image augmentation techniques [27]. Given that the relationship between human body sites and their surrounding environment, such as light shades, hospital beds, and medical appliances, plays a crucial role in classification for our task, we have opted to use Flip and Rotation for data augmentation. These techniques are effective in preserving the relationship between different objects in the image, as opposed to Scale and Crop, which may only retain a portion of the image.

In summary, we have utilized rotation as a means of enhancing data generalization during the training process. Prior to importing images for model training, we randomly rotated them by plus or minus 30 degrees and used bicubic interpolation to complement the rotated image. This is to simulate the potential displacement of the patient’s site during an X-ray and enables the model to better learn the nuanced features of each site. As a result, this approach helps to improve the issue of the small feature gap between the positive and lateral positions, as pointed out by Miao [6].

#### 2.2.3. EfficientNet

Mingxing Tan et al. aimed to develop a model scaling method that could optimize both speed and accuracy. To achieve this, they re-examined several dimensions of model scaling proposed by their predecessors, including network depth, width, and image resolution. While previous studies had typically focused on enlarging one of these dimensions to improve accuracy, the authors recognized that these dimensions are mutually influential and proposed EfficientNet [28] through experiments. Specifically, they first formulated the problem definition to explore the relationship between network depth, width, and image resolution in achieving model accuracy. They assumed that the entire net is *N*, and the *i*-th layer is expressed as: $Y_{i}=F_{i}\left(X_{i}\right)$, where $F_{i}$ is the operator, $Y_{i}$ is output tensor and $X_{i}$ is input tensor. Let *N* consist of *k* convolutional layers, then it can be expressed as: $N=F_{k}⊙\ldots ⊙F_{2}⊙F_{1}\left(X_{1}\right)={⊙}_{j=1\ldots k}F_{j}\left(X_{i}\right).$ In practice, convolutional layers are usually divided into same architecture stages, so *N* can be expressed as follows:(1)$$N={⊙}_{i=1\ldots s}F_{i}^{L_{i}}(X_{\langle H_{i},W_{i},C_{i}\rangle })$$
where *i* is the stage index, and $F_{i}^{L_{i}}$ is the convolutional layer of the *i*-th stage, $F_{i}$ repeats $L_{i}$ times, and $\langle H_{i},W_{i},C_{i}\rangle$ is the shape of the input image.

To reduce the search space, the authors established certain constraints, including fixing the basic structure of the network, imposing equal scaling on all layers, and incorporating memory and computation constraints. As a result, the scaling of the network could only be optimized by multiplying the baseline network defined by ${\hat{F}}_{i}$, ${\hat{L}}_{i}$, ${\hat{H}}_{i}$, ${\hat{W}}_{i}$, ${\hat{C}}_{i}$ in the formula below with a constant magnification:(2)$$\begin{array}{c} \underset{d,w,r}{\max}Accuracy(N(d,w,r))\\ s.t.N\left(d,w,r\right)={⊙}_{i=1\ldots s}{\hat{F}}_{i}^{\hat{L_{i}}}(X_{\langle r\times {\hat{H}}_{i},r\times {\hat{W}}_{i},r\times {\hat{C}}_{i}\rangle })\\ Memory(N)\leq target\_memory\\ FLOPS(N)\leq target\_flops \end{array}$$
where *d*, *w*, *r* are coefficients for scaling network depth, width and resolution.

After conducting experiments that involved adjusting only one dimension at a time, as well as adjusting all three dimensions simultaneously, the authors proposed a compound scaling method. This method involves using a compound coefficient *ϕ* to uniformly scale the network width, depth, and resolution:(3)$$\begin{array}{c} depth:d=\alpha^{\phi }\\ width:w=\beta^{\phi }\\ depth:r=\gamma^{\phi }\\ s.t.\alpha \cdot \beta^{2}\cdot \gamma^{2}\approx 2,where\;\alpha ,\beta ,\gamma \geq 1 \end{array}$$
where α, β, γ are constants that can be determined by a small grid search.

The authors considered that doubling network depth would double FLOPS while doubling network width or resolution would quadruple FLOPS. The FLOPS of a regular operation is proportional to d, w2, r2. As a result, scaling a CNN with Equation (3) would increase the total FLOPS by ${\left(\alpha \cdot \beta^{2}\cdot \gamma^{2}\right)}^{\phi }$. To keep the total FLOPS increase to approximately $2^{\phi }$, they constrained $\alpha \cdot \beta^{2}\cdot \gamma^{2}\approx 2$.

EfficientNet-B0 was generated based on MnasNet [29], and the authors used the compound scaling method to obtain EfficientNet-B1 to EfficientNet-B7. In this study, we chose to use EfficientNet-B3 for our experiments. This is because, compared to B0, B3 increased the accuracy rate by 3.5% with an increase of 6.7 M parameters. In contrast, B4 increased the number of parameters by 7 M compared to B3 but only improved the accuracy rate by 1.3%. Table 2 shows these results. Furthermore, EfficientNet-B3 was found to be a more robust model than Xception [26], which had the best results in Miao’s [6] paper. This is because EfficientNet-B3 showed an improvement of more than 2% over Xception on the ImageNet dataset, as also shown in Table 2.

## 3. Experiments and Results

### 3.1. Experiment Setting

In this study, we employ EfficientNet-B3 to perform image classification on 49 categories of RGB images of the body sites that need to undergo an X-ray examination. We first introduce the dataset and training strategy, followed by the evaluation metrics we use.

**DataSets**: We collected a total of 12,152 images from the X-ray room of the outpatient department at Taichung Veterans General Hospital. We performed data purification to obtain 49 categories and split the data of each category into Training, Validation, and Testing sets at a ratio of 7:1:2. During training, we randomly rotated each image by plus or minus 30 degrees and resized it to 288 × 288. During validation and testing, we resized each image to 320 × 320.

**Training Strategy:** The hyperparameters for this training strategy are as follows: a batch size of 8, Cross-Entropy loss function, Stochastic Gradient Descent (SGD) optimizer with a learning rate of 0.001 and momentum of 0.9. The pre-trained weights from the ImageNet training are used as the initial weights for transfer learning. The training process involves two steps: feature extraction and fine-tuning. In the feature extraction step, we modify the pre-trained model by replacing the fully connected layer to output predictions for the 49 classes in the new task. We then freeze all the layers in the pre-trained model except for the fully connected layer and train only this layer for 10 epochs. This step allows the model to learn to map the features extracted by the pre-trained layers to the new task. In the fine-tuning step, we unfreeze all the layers in the pre-trained model and train the entire model for 30 epochs. This step allows the model to adjust the pre-trained weights to better fit the new task. During this step, the entire model is updated using the SGD optimizer with the specified learning rate and momentum.

**Evaluation Metrics:** The evaluation of the model includes both global and category-specific performance measures, along with a usability assessment. To evaluate the overall performance, we will use the Accuracy metric, which measures the percentage of correctly classified examples in the dataset across all categories. For category-specific performance, the Confusion Matrix will provide a visual representation of the model’s performance in each category. Additionally, Precision, Recall, and F1-score will be calculated for each category to offer a more detailed assessment of performance. These metrics are commonly used in multi-class classification tasks to evaluate the precision (accuracy of positive predictions), recall (sensitivity to true positive examples), and F1-score (harmonic mean of precision and recall) for each category as in [37]. We assessed the values of true positive (TP), false positive (FP), true negative (TN), and false negative (FN) by comparing the ground truth images with the predicted segmented images. The calculations for Accuracy, Precision, Recall, and F1-Score are determined using Equation (4):(4)$$\begin{array}{c} Accuracy=\frac{TP\;+\;TN}{TP\;+\;TN\;+\;FP\;+\;FN}\\ Precision=\frac{TP}{TP\;+\;FP}\\ Recall=\frac{TP}{TP\;+\;FN}\\ F1-Score=\frac{2\;\times \;Precision\;\times \;Recall}{Precision\;+\;Recall} \end{array}$$

### 3.2. Results

#### 3.2.1. EfficientNet

Figure 10 shows the training accuracy and training loss of EfficientNet-B3 on our 49-categories classification task. The training accuracy and training loss serve as crucial indicators to observe the learning progress of the models. By monitoring these metrics during the training process, the programmer can gain insights into how well the model is learning from the data. A high training accuracy and low training loss typically signify that the model is effectively capturing patterns and generalizing well to the training data. In our experiment, after training the model for 30 epochs, we achieved impressive final training accuracy (red line) of 99.89% and validation accuracy (green line) of 97.71% for the 49 categories classification task. The final training loss was 1%, and the validation loss was 8%, indicating good generalization and effective error minimization during training.

#### 3.2.2. Analysis of Results

From Table 3 and Table 4, it is evident that L’t ELBOW LAT is the site with the lowest Recall and F1-score and also makes the most mistakes (eight times) in all categories, as shown in Figure 11. It is also observed that all categories with the lowest F1-score are from three sites, ELBOW, FEMUR, and LEG. The categories of FEMUR and LEG cover a large area, making it more challenging for the model to understand the complex relationship between the site and the surrounding environment to classify the different directions accurately. On the other hand, ELBOW images did not capture the real difference between the categories in different directions, and the categories in different directions look very similar in appearance, as shown in Figure 12.

To address these challenges, the proposed two-stage improvement approach aims to capture subtle feature differences between individual sites more accurately, particularly for the difficult-to-classify sites. The proposed hierarchical classification approach is expected to improve the accuracy by more accurately capturing the feature differences of these challenging sites.

#### 3.2.3. Two-Stage Improvement

T40P3x4

In the proposed two-stage improvement approach, T40P3x4, the first stage involves merging the 12 categories derived from the 3 sites with the lowest F1-score into 3 categories: ELBOW, FEMUR, and LEG, thereby reducing the total categories to 40. In the second stage, a 4-categories classifier is trained for each of the three merged categories to classify them back to the original category.

The proposed architecture, T40P3x4, consists of two stages, and EfficientNet-B3 is used for the model in any stage. Compared to the previous classifier, S49 (Single stage: 49-categories), this new architecture is expected to improve the accuracy of the model by capturing subtle feature differences between individual sites more accurately, as shown in Figure 13.

The training strategy of the first stage model of T40P3x4 is similar to that of training S49, with the exception of some modifications in the data set distribution. The S49 data set is used as the basis for modification, but due to the merging of multiple categories in the current classification, directly aggregating the data sets of multiple categories into one new category may cause data imbalance. Therefore, for the three sites merged into one category, we controlled the total number of data for the new category at 300. We performed uniform sampling on the Training set, Validation set, and Testing set for the four categories to be merged into a new category on average. So, the subcategories of these 3 sites will all contribute 53, 7, and 15 images to their new category in sequence, totaling 75 images. Please refer to Table 5 for more details.

Figure 14 shows the training accuracy and training loss of the first-stage model of T40P3x4. Table 6 show the data distribution of the second stage input image. The final training accuracy was 99.91%, training loss was 1%, validation accuracy was 98.94%, validation loss was 5%, and the testing accuracy was 98.82%. In terms of the training results of the classifiers in the second stage, all classifiers had a training accuracy of more than 98.5%, as shown in Table 7. Regarding testing accuracy, ELBOW achieved 94.12%, FEMUR achieved 94.95%, and LEG achieved 87.38%. Combining these two stages’ classifiers, we obtained an overall testing accuracy of 97.99% on the original 49 categories. The confusion matrix is shown in Figure 15. In the following section, we will compare the training results of T40P3x4 and S49.

#### 3.2.4. Three-Stage Improvement

T40P2x4A2P2

In Table 7, it is evident that T40P3x4 is not suitable for LEG classification. The confusion matrix in Figure 15 indicates a higher probability of misclassifying R’t LEG AP, R’t LEG LAT, and L’t LEG AP. As a result, we propose an alternative architecture, T40P2x4A2P2, as shown in Figure 16. In this architecture, we have divided the LEG parts into two stages. We first classify the AP and LAT as LEG AP/LAT and then classify the R’t and L’t as LEG R’t/L’t.

In the training strategy we have used the same training strategy as T40P3x4. Figure 17 displays the training accuracy and training loss of the first stage model of T40P2x4A2P2, and Table 8 shows the final training results. The Training accuracy, Training Loss, and Testing Accuracy of the LEG AP classifier are 98.32%, 4%, and 98.08%, respectively. The Training accuracy, Training Loss, and Testing Accuracy of the LEG RL classifier are 97.60%, 4%, and 92.31%, respectively. The testing accuracy for ELBOW, FEMUR, and LEG are 94.12%, 94.95%, and 93.16%, respectively.

When we replaced the two-stage LEG classifiers with the original T40P3x4 LEG classifiers, we achieved an overall testing accuracy of 98.16% on the original 49 categories, as shown in the confusion matrix in Figure 18. In the following section, we will compare the training results of T40P3x4 and S49.

## 4. Discussion

In this section, we will discuss the results and compare them with those of S49, and also explore possibilities for future work.

Results Comparison

Table 9 shows the comparison of 12 categories in terms of F1-score between S49, T40P3x4, and T40P2x4A2P2. The lower F1-score of T40P3x4 in the LEG categories can be attributed to the fact that the model’s understanding of LEG features may have originated from the features learned in other body parts. Therefore, when training a classifier solely related to LEG-derived categories, the model may fail to achieve effective learning outcomes.

However, it is important to note that although T40P3x4 enhances the overall accuracy from 97.59% to 98.00%, the performance of LEG does not improve and instead declines. On the other hand, T40P2x4A2P2 improves the F1-score of LEG-derived categories and, when using the same classifier as T40P3x4, can enhance the overall accuracy from 98.00% to 98.16%.

Table 10 displays the results of the comparison between our three proposed model architectures and the four models used in [6]. The model architecture employed in [6] directly classifies 49 categories, making it identical to our S49 model. However, our three model architectures achieved higher testing accuracy compared to the four models used in [6]. This outcome suggests that our proposed models outperformed the models utilized in [6] when it comes to the accurate classification of the 49 categories.

Future Work

Table 9 emphasizes the enhancements in F1-score achieved by three different model architectures, while Table 10 showcases the testing accuracy of these models. To further improve both F1 scores and classification accuracy, it is vital to consider employing diverse model architectures or even exploring alternative models like DenseNet. As a future direction, we propose conducting thorough investigations into various model architectures to identify the most suitable one for augmenting the overall classification performance. This approach holds the potential to yield more accurate and dependable results, thereby enhancing the efficiency and effectiveness of the classification system.

## 5. Conclusions

Patient safety is crucial in medicine, and reducing delayed diagnoses is vital for achieving this goal. We developed a medical assistance system for X-ray inspections using deep learning techniques and data from Taichung Veterans General Hospital’s radiology department. The system effectively minimizes errors and receives positive feedback from users. By implementing this system, we enhance medical processes and improve the quality of services while prioritizing patient safety. The reduction in delayed diagnoses prevents potential harm and fosters a safer medical environment.

The primary focus of this study was to address the multi-classification task in X-ray inspections. To achieve this, we employed EfficientNet [26] for training and testing and introduced data purification and augmentation enhancements customized for the task. Additionally, we proposed two different architectures to further enhance the classification accuracy. As a result of directly implementing EfficientNet into the classifier, the system’s accuracy increased significantly from 94.10% (as in [6]) to 97.59%. Subsequent experiments involved fine-tuning the system architecture and adopting two-stage and three-stage classification approaches, resulting in impressive overall accuracies of 98% and 98.16%, respectively. To further improve classification accuracy, future research should explore new model architectures or different neural networks. These efforts hold the potential to elevate the accuracy even further, advancing the effectiveness of the classification system.

## Figures and Tables

**Figure 1 healthcare-11-02068-f001:**
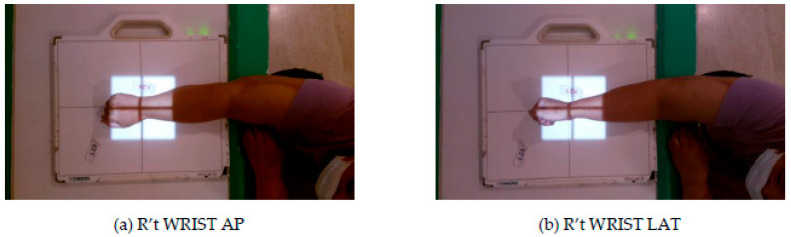
Example of Irradiating the Wrong Direction.

**Figure 2 healthcare-11-02068-f002:**
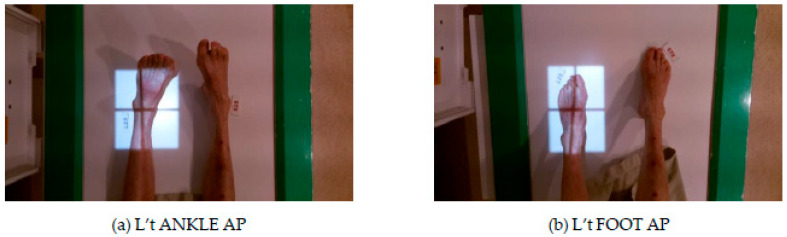
Example of Irradiating the Wrong Site.

**Figure 3 healthcare-11-02068-f003:**
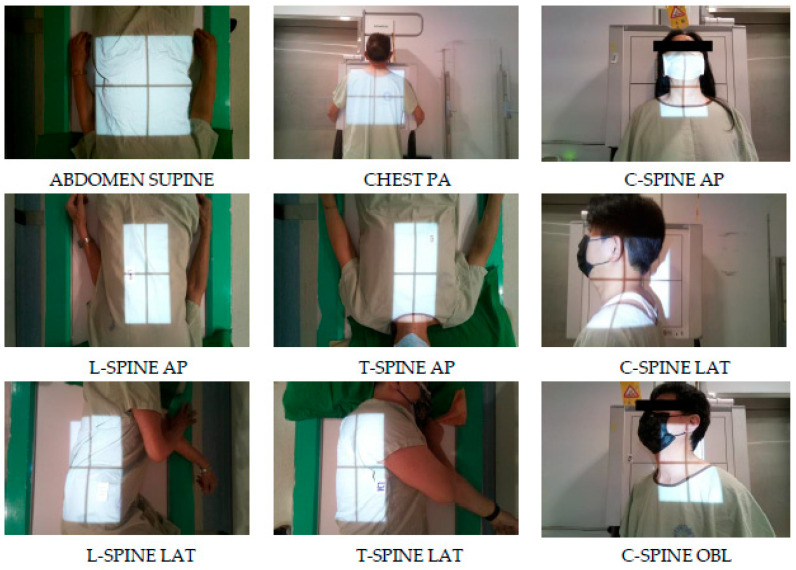
Non-Directional sites.

**Figure 4 healthcare-11-02068-f004:**
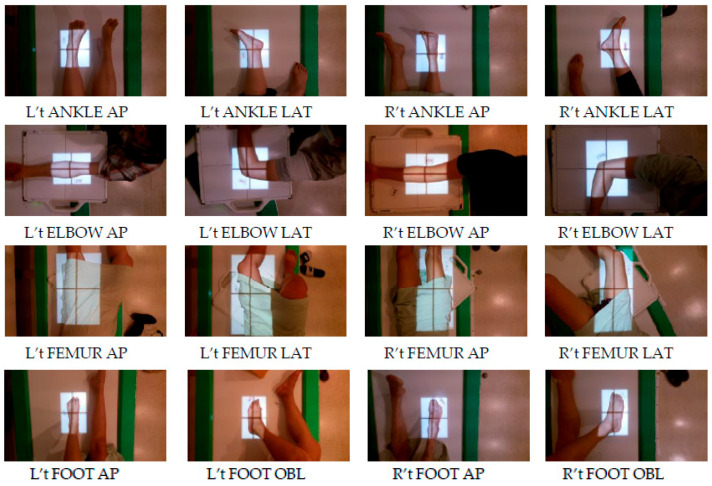
Directional sites.

**Figure 5 healthcare-11-02068-f005:**
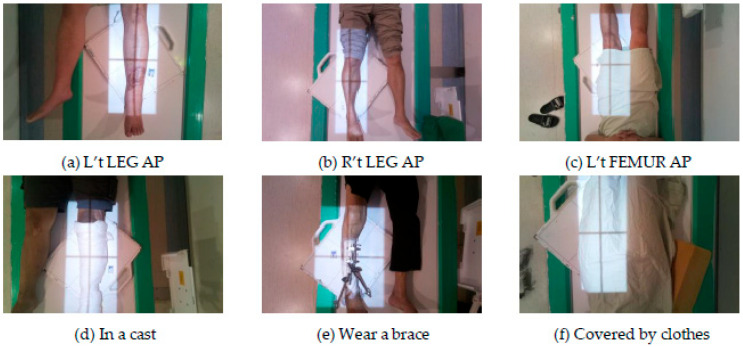
Changes in the appearance of the irradiated site. The **upper** row is the most common way to irradiate that site, and the **lower** row is the special situation.

**Figure 6 healthcare-11-02068-f006:**
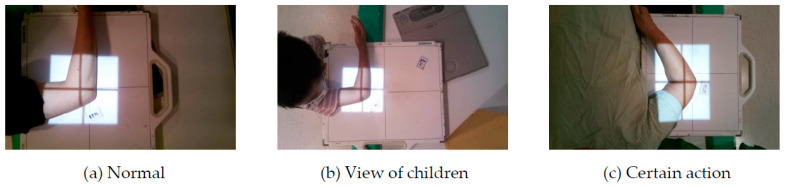
Changes in the placement of the irradiated site. (**a**) is the most common way to irradiate R’t ELBOW LAT, and the other two images (**b**,**c**) are the special situation.

**Figure 7 healthcare-11-02068-f007:**
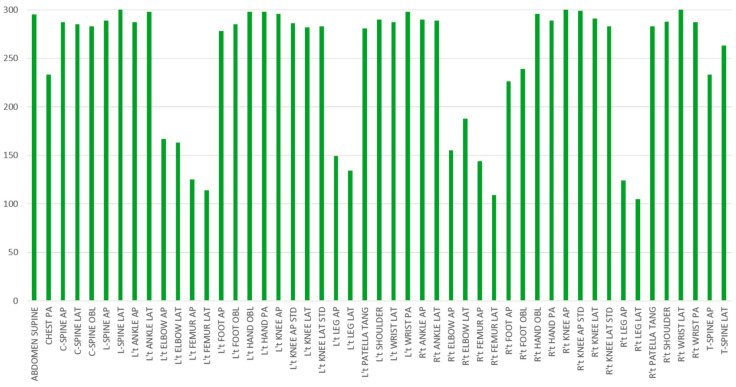
The distribution of the number of images in each category.

**Figure 8 healthcare-11-02068-f008:**
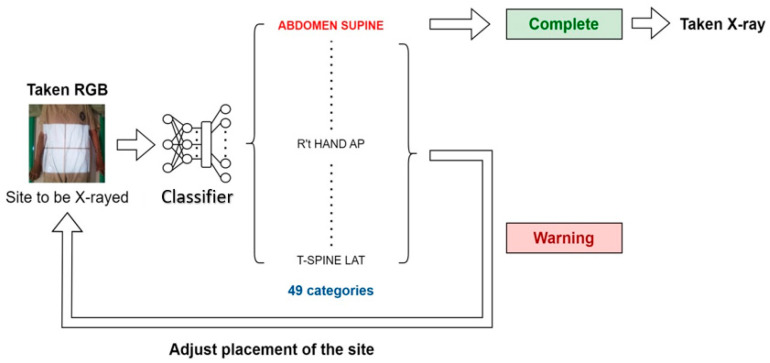
System Workflow.

**Figure 9 healthcare-11-02068-f009:**
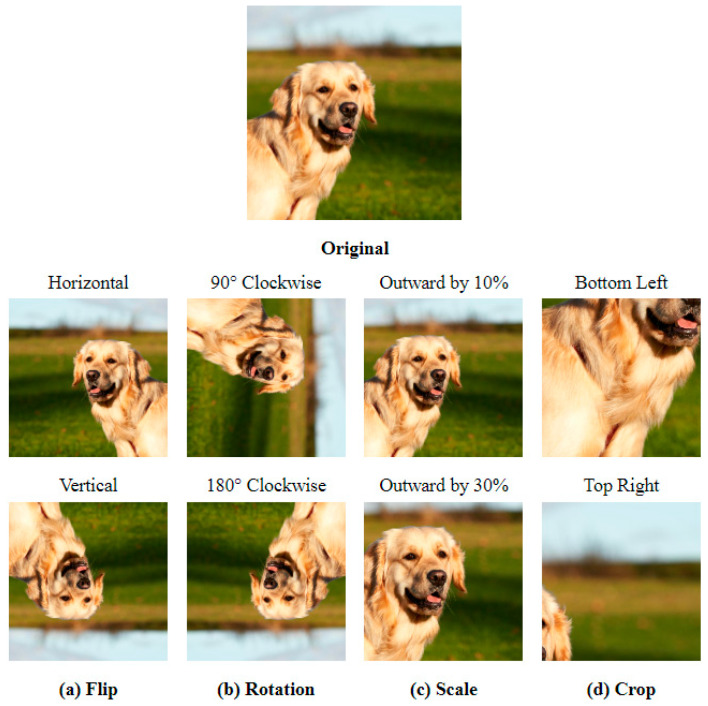
General Augmentation Techniques. This image is from Pexels https://www.pexels.com/photo/close-up-shot-of-a-golden-retriever-10096129/).

**Figure 10 healthcare-11-02068-f010:**
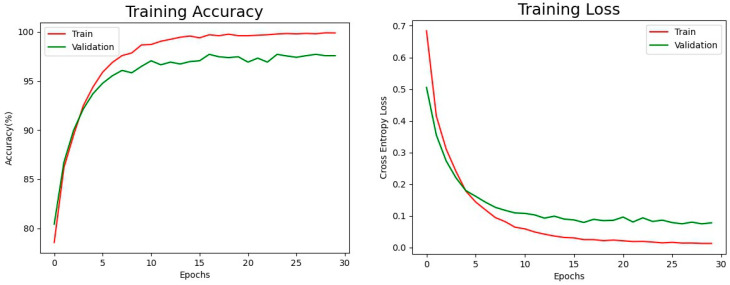
The Training Results of EfficientNet-B3.

**Figure 11 healthcare-11-02068-f011:**
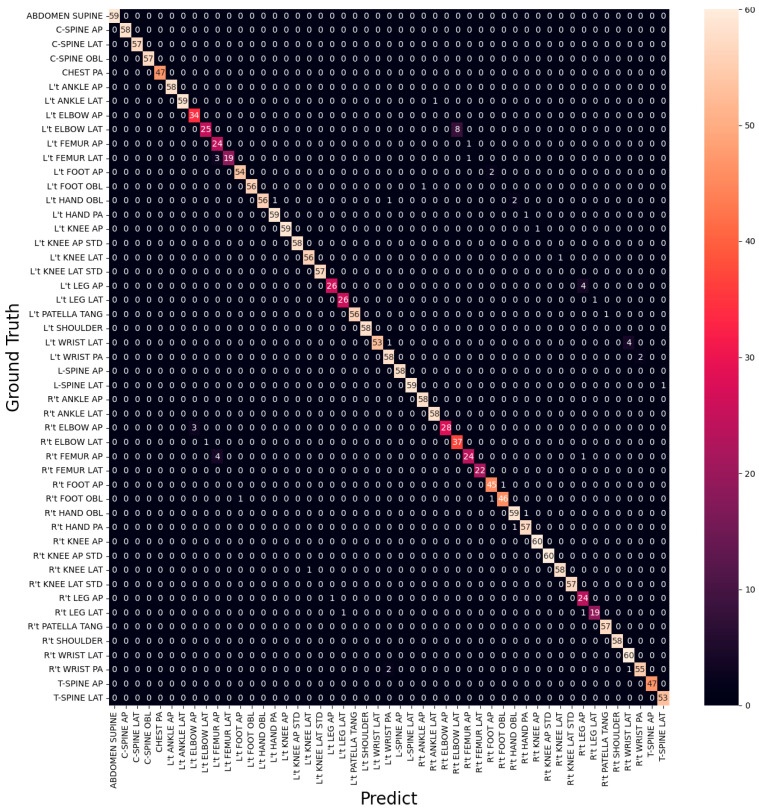
Confusion Matrix of EfficientNet-B3.

**Figure 12 healthcare-11-02068-f012:**
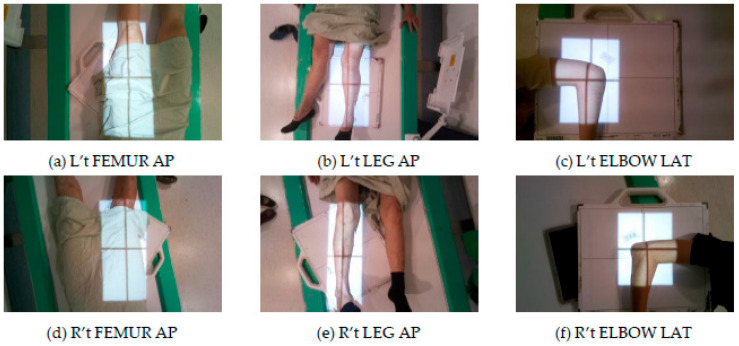
Sites that are difficult to classify.

**Figure 13 healthcare-11-02068-f013:**
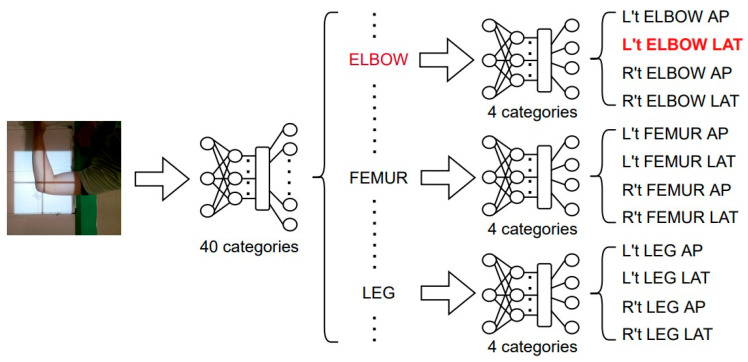
Architecture of T40P3x4. The red color words means the input image will go through the path.

**Figure 14 healthcare-11-02068-f014:**
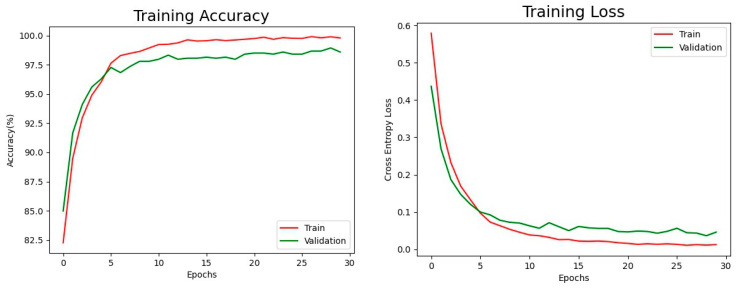
Training Results of the first stage model of T40P3x4.

**Figure 15 healthcare-11-02068-f015:**
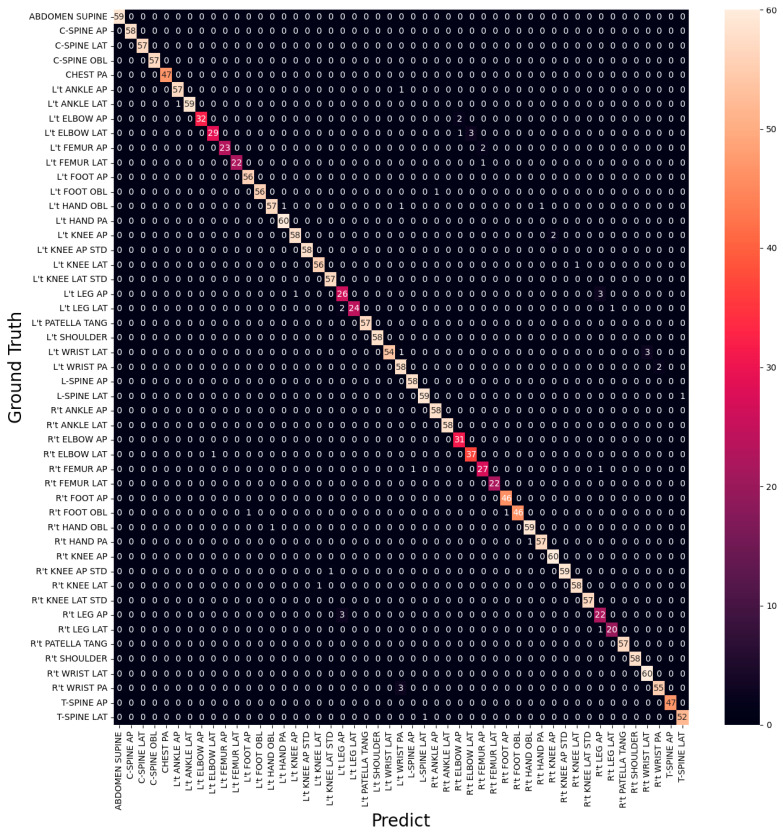
Confusion Matrix on 49 categories of T40P3x4.

**Figure 16 healthcare-11-02068-f016:**
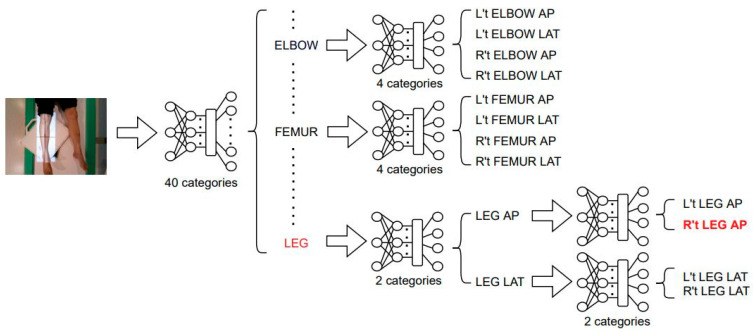
Architecture of T40P2x4A2P2. The red color words means the input image will go through the path.

**Figure 17 healthcare-11-02068-f017:**
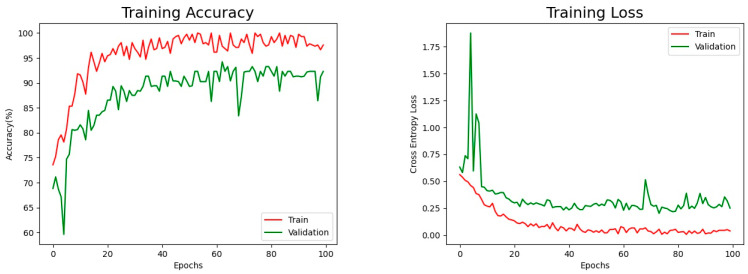
Training Results of the first stage model of T40P2x4A2P2.

**Figure 18 healthcare-11-02068-f018:**
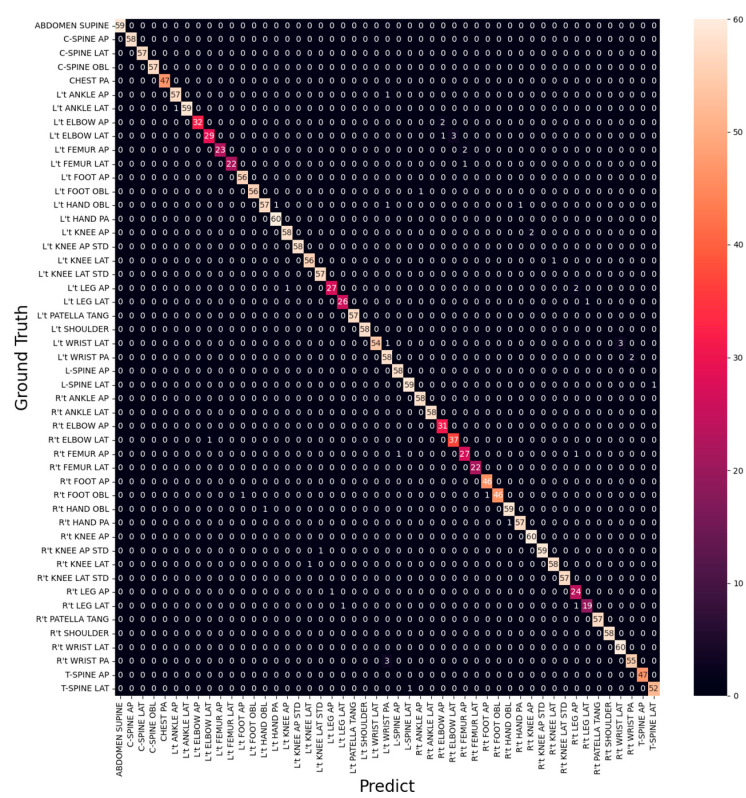
Confusion Matrix on 49 categories of T40P2x4A2P2.

**Table 1 healthcare-11-02068-t001:** The number of images in each category.

Category	L’t			R’t			Non-Direc	Sum
	AP/PA	LAT	OBL	AP/PA	LAT	OBL		
ABDOMENSUPINE	x	x	x	x	x	x	295	295
CHESTPA	x	x	x	x	x	x	233	233
C-SPINEAP	x	x	x	x	x	x	287	287
C-SPINELAT	x	x	x	x	x	x	285	285
C-SPINEOBL	x	x	x	x	x	x	283	283
L-SPINEAP	x	x	x	x	x	x	289	289
L-SPINELAT	x	x	x	x	x	x	300	300
T-SPINEAP	x	x	x	x	x	x	233	233
T-SPINELAT	x	x	x	x	x	x	263	263
ANKLE	287	298	x	290	289	x	x	1164
ELBOW	167	163	x	155	188	x	x	673
FEMUR	125	114	x	144	109	x	x	492
FOOT	278	x	285	226	x	239	x	1028
HAND	298	x	298	289	x	296	x	1181
KNEE	296	282	x	300	291	x	x	1169
KNEESTD	286	283	x	299	283	x	x	1151
LEG	149	134	x	124	105	x	x	512
PATELLATANG	281	x	x	283	x	x	x	564
SHOULDER	290	x	x	288	x	x	x	578
WRIST	298	287	x	287	300	x	x	1172
Total	2755	1561	583	2685	1565	2468	2468	12,152

“x” means no case.

**Table 2 healthcare-11-02068-t002:** EfficientNet Performance Results on ImageNet. The content is from Table 2 of [28].

Model	Top-1 Acc	Top-5 Acc	#Params	Ratio-to-EfficientNet	#FLPs	Ratio-to-EfficientNet
EfficientNet-B0	77.1%	93.3%	5.3 M	1×	0.39 B	1×
ResNet-50 [18]	76.0%	93.0%	26 M	4.9×	4.1 B	11×
DenseNet-169 [20]	76.2%	93.2%	14 M	2.6×	3.5 B	8.9×
EfficientNet-B1	79.1%	94.4%	7.8 M	1×	0.70 B	1×
ResNet-152 [18]	77.8%	93.8%	60 M	7.6×	11 B	16×
DenseNet-264 [20]	77.9%	93.9%	34 M	4.3×	6.0 B	8.6×
Inception-v3 [16]	78.8%	94.4%	24 M	3.0×	5.7 B	8.1×
Xception [26]	79.0%	94.5%	23 M	3.0×	8.4 B	12×
EfficientNet-B2	80.1%	94.9%	9.2 M	1×	1.0 B	1×
Inception-v4 [17]	80.0%	95.0%	48 M	5.2×	13 B	13×
Inception-resnet-v2 [17]	80.1%	95.1%	56 M	6.1×	13 B	13×
EfficientNet-B3	81.6%	95.7%	12 M	1×	1.8 B	1×
ResNeXt-101 [30]	80.9%	95.6%	84 M	7.0×	32 B	18×
PolyNet [31]	81.3%	95.8%	92 M	7.7×	35 B	19×
EfficientNet-B4	82.9%	96.4%	19 M	1×	4.2 B	1×
SENet [21]	82.7%	96.2%	146 M	7.7×	42 B	10×
NASNet-A [32]	82.7%	96.2%	89 M	4.7×	24 B	5.7×
AmoebaNet-A [33]	82.8%	96.1%	87 M	4.6×	23 B	5.5×
PNASNet [34]	82.9%	96.2%	86 M	4.5×	23 B	6.0×
EfficientNet-B5	83.6%	96.7%	30 M	1×	9.9 B	1×
AmoebaNet-C [35]	83.5%	96.5%	155 M	5.2×	41 B	4.1×
EfficientNet-B6	84.0%	96.8%	43 M	1×	19 B	1×
EfficientNet-B7	84.3%	97.0%	66 M	1×	37 B	1×
Gpipe [36]	84.3%	97.0%	557 M	8.4×	-	-

**Table 3 healthcare-11-02068-t003:** Categories with Recall lower than 90%.

Category	Precision (%)	Recall (%)	F1-Score (%)	Support
L’t ELBOW LAT	96.15	75.76	84.75	33
L’t FEMUR LAT	100.00	82.61	90.48	23
R’t FEMUR AP	92.31	82.76	87.27	29
L’t LEG AP	96.30	86.67	91.23	30

**Table 4 healthcare-11-02068-t004:** Categories with F1-score lower than 95%.

Category	Precision (%)	Recall (%)	F1-Score (%)	Support
L’t ELBOW LAT	96.15	75.76	84.75	33
L’t FEMUR AP	77.42	96.00	85.71	25
R’t FEMUR AP	92.31	82.76	87.27	29
R’t LEG AP	80.00	96.00	87.27	25
R’t ELBOW LAT	82.22	97.37	89.16	38
L’t FEMUR LAT	100.00	82.61	90.48	23
L’t LEG AP	96.30	86.67	91.23	30
R’t LEG LAT	95.00	90.48	92.68	21
R’t ELBOW AP	100.00	90.32	94.92	31

**Table 5 healthcare-11-02068-t005:** Data distribution of merged categories for the first stage model of T40P3x4. The right number represents the number of data of that original category in S49.

New Category	Original Category	Training Set	Validation Set	Testing Set	Sum	Total
ELBOW	L’t ELBOW AP	53/116	7/17	15/34	75	300
	L’t ELBOW LAT	53/114	7/16	15/33	75	
	R’t ELBOW AP	53/108	7/16	15/31	75	
	R’t ELBOW LAT	53/131	7/19	15/38	75	
FEMUR	L’t FEMUR AP	53/87	7/13	15/25	75	300
	L’t FEMUR LAT	53/79	7/12	15/23	75	
	R’t FEMUR AP	53/100	7/15	15/29	75	
	R’t FEMUR LAT	53/76	7/11	15/22	75	
LEG	L’t LEG AP	53/104	7/15	15/30	75	300
	L’t LEG LAT	53/93	7/14	15/27	75	
	R’t LEG AP	53/86	7/13	15/25	75	
	R’t LEG LAT	53/73	7/11	15/21	75	

**Table 6 healthcare-11-02068-t006:** Data distribution of different classifiers in the second stage of T40P3x4.

Classifier	Category	Training Set	Validation Set	Testing Set	Sum	Total
ELBOW	L’t ELBOW AP	113	34	34	167	673
	L’t ELBOW LAT	130	33	33	163	
	R’t ELBOW AP	124	31	31	155	
	R’t ELBOW LAT	150	38	38	188	
FEMUR	L’t FEMUR AP	100	25	25	125	492
	L’t FEMUR LAT	91	23	23	114	
	R’t FEMUR AP	115	29	29	144	
	R’t FEMUR LAT	87	22	22	109	
LEG	L’t LEG AP	119	30	30	149	512
	L’t LEG LAT	107	27	27	134	
	R’t LEG AP	99	25	25	124	
	R’t LEG LAT	84	21	21	105	

**Table 7 healthcare-11-02068-t007:** Training results of classifiers in the second stage of T40P3x4.

Classifier	Training Accuracy (%)	Training Loss (%)	Testing Accuracy (%)
ELBOW	99.82	5	94.12
FEMUR	98.75	10	94.95
LEG	99.04	10	87.38

**Table 8 healthcare-11-02068-t008:** Training results of classifiers in the second stage and third stage of T40P2x4A2P2.

Classifier	Training Accuracy (%)	Training Loss (%)	Testing Accuracy (%)
ELBOW	99.82	5	94.12
FEMUR	98.75	10	94.95
LEG AP/LAT	98.32	4	98.08
LEG R’t/L’t	97.60	4	92.31

**Table 9 healthcare-11-02068-t009:** Comparison F1-score (%) on the ELBOW, FEMUR and LEG in S49, T40P3x4 and T40P2x4A2P2. The bold represents the F1-score is lower than 90%.

Category		S49	T40P3x4	T40P2x4A2P2
ELBOW	L’t ELBOW AP	95.77	96.97 (+1.20)	96.97 (+1.20)
	L’t ELBOW LAT	**84.75**	92.06 (+7.31)	92.06 (+7.31)
	R’t ELBOW AP	94.92	95.39 (+0.47)	95.39 (+0.47)
	R’t ELBOW LAT	**89.16**	94.87 (+5.71)	94.87 (+5.71)
	Overall	91.15	94.82	94.82
FEMUR	L’t FEMUR AP	**85.71**	95.83 (+10.12)	95.83 (+10.12)
	L’t FEMUR LAT	90.48	97.78 (+7.30)	97.78 (+7.30)
	R’t FEMUR AP	**87.27**	91.53 (+4.26)	91.53 (+4.26)
	R’t FEMUR LAT	100.00	100.00	100.00
	Overall	90.86	96.28	96.28
LEG	L’t LEG AP	91.23	**85.25 (−5.98)**	93.10 (+1.87)
	L’t LEG LAT	96.30	94.12 (−2.18)	96.30
	R’t LEG AP	**87.27**	**84.62 (−2.65)**	90.57 (+3.30)
	R’t LEG LAT	92.68	95.24 (+2.56)	92.68
	Overall	91.87	**89.81**	93.16
Overall		97.59	98.00	98.16

**Table 10 healthcare-11-02068-t010:** Comparison of Three Proposed Model Architectures with Four Models Used in [6].

Model	Miao [6]				Our Methods		
	Xception	Inception V3	ResNet50	VGG16	S49	T40P3x4	T40P2x4A2P2
Testing accuracy (%)	94.10	92.00	92.50	90.50	97.59	98.00	98.16

## Data Availability

Not applicable.

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
