# Peer review of "Improving Patient Safety in the X-ray Inspection Process with EfficientNet-Based Medical Assistance System"

_healthcare, 2023, doi:10.3390/healthcare11142068_

Round 1

Reviewer 1 Report

In this study, the authors propose an EfficientNet multi-classification method for medical assistance that will reduce X-ray inspection errors. A background, a design architecture, as well as satisfactory experimental results are included in the manuscript. The novelty of the paper, however, is incremental, and there are no new ideas or applications presented in the paper.

Additional comments:

An abstract should be concise, and convey the relevance of the problem, the design architecture, and the results in as few words as possible.

The sentence starting with "The latest Taiwan Patient Safety Reporting system ..." should be placed in Introduction, not in abstract.

Lines 74 - 221. CNN's history is not necessary for introduction, so users may move it to another subsection, history or background if necessary. Figures related to them may also be moved to that section. In my opinion, however, these details are not relevant to the subject of this paper.

Both Table 1 and Figure 12 provide the same information. Keep both if they are not redundant, otherwise remove one of them.

There are some parts of the manuscript that are not written in English. It is impossible for me to evaluate even when the results section contains unreadable words.

Space is missing in many places. 

Paper is not fully written in English.

Figure 18. Caption is not readable.

Author Response

We have followed your suggestions modified our manuscripts as the attached file. 

Reviewer 2 Report

In this paper, authors proposed two enhanced Neural Network architectures to improve the medical operation process during X-ray examination. They capture an RGB image of the patient's site before X-ray irradiation and use a Convolutional Neural Network (CNN) model for image classification.  Authors did a good work and interested for the readers. Following review comments are recommended, and the authors are invited to explain and modify.

1 What is justification of Title: “Medical Assistance System on X-Ray Inspection Process with EfficientNet”?

2 Introduction section is unnecessarily wordy. Make it brief and concise.

3 Novelty is confusing. A highlight is required. The main contributions of the manuscript are not clear. The main contributions of the ‎article must be very clear and would be better if summarize ‎them into 3-4 points at the ‎end of the introduction.‎

4 Methodology section should have a detailed flowchart of the whole work. This will help the reader to get a better understanding of what is going on in the proposed ‎system.‎

5 Introduction section needs to be improve with latest references, no any ref was found of years 2022 and 2023. An introduction is an important road map for the rest of the paper that should be consist of an opening hook to catch the researcher's attention, relevant background study, and a concrete statement that presents main argument but your introduction lacks these fundamentals, especially relevant background studies. This related work is just listed out without comparing the relationship between this paper's model and them; only the method flow is introduced at the end; and the principle of the method is not explained. To make soundness of your study must include these latest related works. Authors also need to justify the importance of their article and cite all of them to make a critical discussion that makes a difference from others' work.

I (2023). iERM: An Interpretable Deep Learning System to Classify Epiretinal Membrane for Different Optical Coherence Tomography Devices: A Multi-Center Analysis. Journal of Clinical Medicine, 12(2). doi: 10.3390/jcm12020400

II (2022). Automatic interpretation and clinical evaluation for fundus fluorescein angiography images of diabetic retinopathy patients by deep learning. British Journal of Ophthalmology, 2022-321472. doi: 10.1136/bjo-2022-321472

III (2023). MSHF: A Multi-Source Heterogeneous Fundus (MSHF) Dataset for Image Quality Assessment. Scientific Data, 10(1), 286. doi: 10.1038/s41597-023-02188-x

IV (2021). Is Health Contagious?—Based on Empirical Evidence From China Family Panel Studies' Data. Frontiers in Public Health, 9. doi: 10.3389/fpubh.2021.691746

6 “The hyperparameters for this training strategy”, how to optimize hyperparameters during model training?

7 Why did it chose EfficientNet-B3 to perform image classification?

8 When writing phrases like “Precision, Recall, and F1-score will be calculated for each category to provide a more detailed assessment of performance”, it must cite related work in order to sustain the statement https://doi.org/10.1155/2023/2345835

9 Figures 14, 18, and 21 require detailed description.

10 Authors should mention the implementation challenges.

Moderate editing of English language required.

Author Response

We have followed your suggestions to modified our manuscript as the attached file.

Reviewer 3 Report

Title: "Improving Patient Safety in the X-Ray Inspection Process with EfficientNet-based Medical Assistance System"

Abstract:

The abstract provides a concise summary of the study, highlighting the importance of patient safety in the medical field and the need for improved processes. The utilization of deep learning techniques, specifically EfficientNet, is mentioned as a means to enhance the accuracy of the X-ray inspection process. The study claims to have achieved a significant improvement in accuracy compared to previous research. 

Suggestions:

1. Introduction:

a. Provide a more detailed background on the current state of patient safety in the medical field. Include relevant statistics or studies to emphasize the significance of inspection incidents and their impact on patient health.

b. Clarify the specific challenges and risks associated with the X-ray inspection process that the proposed system aims to address. This will help readers understand the context and motivation for the study.

2. Methodology:

a. Clearly describe the dataset used from Taichung Veterans General Hospital. Include information on the number of patients, the range of conditions represented, and any potential biases or limitations.

b. Provide an overview of the EfficientNet architecture and explain why it was chosen for this study. Discuss any modifications or adaptations made to the network for the specific task of X-ray image classification.

c. Elaborate on the data purification and augmentation techniques used. Explain how these techniques addressed the specific challenges and characteristics of the X-ray inspection process.

3. Results and Discussion:

a. Present a comprehensive analysis of the experimental results. Include performance metrics such as precision, recall, and F1-score to provide a more detailed evaluation of the system's effectiveness.

b. Discuss the limitations and potential sources of error in the system. Address any challenges encountered during the implementation or evaluation phases and propose potential solutions or future directions for improvement.

c. Compare the achieved accuracy with existing state-of-the-art methods or studies in the field to provide a more robust assessment of the system's performance.

4. Conclusion:

a. Emphasize the impact of the proposed medical assistance system on patient safety and its potential to reduce errors in the X-ray inspection process.

b. Summarize the key findings and contributions of the study concisely.

c. Discuss the broader implications of the research and highlight areas for future research or implementation.

5. References:

a. Ensure that all references are up-to-date and relevant to the topic.

b. Include a mix of recent studies and seminal works in the field to support the claims made in the paper.

Overall, the paper addresses a significant issue in the medical field and presents a promising solution utilizing deep learning techniques. By incorporating the suggested revisions, the paper will provide a more comprehensive understanding of the research, its methodology, and the implications for improving patient safety in the X-ray inspection process.

English presentation should be improved.

Author Response

(The authors gave the same response as above.)

Reviewer 4 Report

Overall, your paper presents a valuable contribution to the field and addresses an important aspect of medical technology. With some minor revisions and additions, I believe your paper will be even stronger and more impactful.

The overall language usage and grammar are sound, and the message is effectively conveyed. However, there is room for improvement in terms of providing more specific details and enhancing the clarity and flow of the text in certain sections. Nonetheless, the editing is of a satisfactory standard.

Author Response

We deeply appreciate your prompt response and invaluable feedback. We have made the necessary modifications. Below, we have provided the modified lines in our manuscript:

Lines 27-35

Lines 42-44.

Lines 70-82.

Lines 103-117.

Lines 155-160.

Lines 174-181.

Lines 288-297.

Lines 418-424.

Lines 441-451.

Round 2

Reviewer 1 Report

The authors have accommodated all the suggested changes. 

Author Response

We deeply appreciate your prompt response and invaluable feedback. We have made the necessary modifications.

Reviewer 2 Report

We appreciated the authors' efforts in manuscript revision. However, following concerns need to be discussed and revised carefully before the paper's acceptance.

1 Table 1 and Figure 7 have same purpose so better to delete one.

2 “Figure 8. System Workflow”, needs detailed description of whole method.

3 Introduction section needs to be improve with latest references, no any reference was found of years 2022 and 2023, but still authors did not cite latest related references and discuss them.   

I (2023). iERM: An Interpretable Deep Learning System to Classify Epiretinal Membrane for Different Optical Coherence Tomography Devices: A Multi-Center Analysis. Journal of Clinical Medicine, 12(2). doi: 10.3390/jcm12020400

II (2023). MSHF: A Multi-Source Heterogeneous Fundus (MSHF) Dataset for Image Quality Assessment. Scientific Data, 10(1), 286. doi: 10.1038/s41597-023-02188-x

III (2021). Is Health Contagious?—Based on Empirical Evidence From China Family Panel Studies' Data. Frontiers in Public Health, 9. doi: 10.3389/fpubh.2021.691746

4 “Figure 9. Popular Augmentation Techniques. This image is from Pexels”, what did it need to use “Popular”?

5 It is not need to show “Training Accuracy (%)” in Tables.  

6 In response to writing phrases like “Precision, Recall, and F1-score will be calculated for each category to provide a more detailed assessment of performance”, it must cite related work in order to sustain the statement 10.1155/2023/2345835; 10.1117/12.2540175.

“We have taken inspiration from the research presented”, but authors still missed to include these references.

Minor editing of English language required.

Author Response

We have modified our manuscript as the attached file.

Reviewer 3 Report

I appreciate the effort made by authors in addressing my observations. The manuscript has improved and the authors managed to address my questions. In my view, the paper can be accept.

The article should be checked for language presentation for the last time.

Author Response

(The authors gave the same response as above.)
